# Electrospun Asymmetric Membranes as Promising Wound Dressings: A Review

**DOI:** 10.3390/pharmaceutics13020183

**Published:** 2021-01-30

**Authors:** Mariana F. P. Graça, Duarte de Melo-Diogo, Ilídio J. Correia, André F. Moreira

**Affiliations:** 1CICS-UBI—Centro de Investigação em Ciências da Saúde, Universidade da Beira Interior, Av. Infante D. Henrique, 6200-506 Covilha, Portugal; mariana.graca@ubi.pt (M.F.P.G.); demelodiogo@fcsaude.ubi.pt (D.d.M.-D.); 2CIEPQPF—Departamento de Engenharia Química, Universidade de Coimbra, Rua Silvio Lima, 3030-790 Coimbra, Portugal

**Keywords:** asymmetric membranes, bioactive molecules, electrospun membranes, skin regeneration, wound dressing

## Abstract

Despite all the efforts that have been done up to now, the currently available wound dressings are still unable to fully re-establish all the structural and functional properties of the native skin. To overcome this situation, researchers from the tissue engineering area have been developing new wound dressings (hydrogels, films, sponges, membranes) aiming to mimic all the features of native skin. Among them, asymmetric membranes emerged as a promising solution since they reproduce both epidermal and dermal skin layers. Wet or dry/wet phase inversion, scCO_2_-assisted phase inversion, and electrospinning have been the most used techniques to produce such a type of membranes. Among them, the electrospinning technique, due to its versatility, allows the development of multifunctional dressings, using natural and/or synthetic polymers, which resemble the extracellular matrix of native skin as well as address the specific requirements of each skin layer. Moreover, various therapeutic or antimicrobial agents have been loaded within nanofibers to further improve the wound healing performance of these membranes. This review article provides an overview of the application of asymmetric electrospun membranes as wound dressings displaying antibacterial activity and as delivery systems of biomolecules that act as wound healing enhancers.

## 1. Introduction

Skin is the largest and outermost organ of the human body, with approximately 2 m^2^ of area and a mean thickness of 2.5 mm [1,2]. This organ is involved in important functions in the human body, namely thermoregulation, prevention of water and fluid loss, immune surveillance, hormone synthesis, and sensory detection [1,3]. In addition, due to its anatomical location, it also acts as a barrier against microbial invasion as well as mechanical and chemical insults, thus conferring protection to the body [3]. In this way, when the skin’s structure is compromised, the use of dressing materials to cover and protect the wound to re-establish a temporary or, in the case of extensive wounds, a permanent fully functional body barrier is of utmost importance [4].

In this field, various biomedical alternatives have been developed and applied over the years to assist the wound healing process. The skin grafts (autografts, allografts, xenografts) remain as the most conventional and widely used therapeutic approach for restoring the skin’s structure after an extensive lesion [5]. Despite their intrinsic advantages, autografts present limited availability and induce additional morbidity to the patient; while alo- and xenografts can lead to immune rejection [6,7]. In turn, wound dressings, i.e., 3-dimensional materials/structures that can be applied in the wound site either temporarily or permanently, act as a barrier against microorganisms, external insults, and dehydration, while simultaneously accelerating the wound healing [3,8]. Today, some wound dressings are already applied in the clinic, for example, Duoderm^®^ [9], Acticoat^™^ [10], Aquacel Ag^®^ [11], DermFactor^®^ [12], and Procellera^®^ [13]. Despite the advantages presented by these commercial wound dressings, they still present drawbacks such as adhesion to the surface of the lesion, which may cause additional damage upon periodic replacement, and the cost [8].

To address these limitations, tissue engineering researchers have been focusing on the development of new different biomimetic wound dressings. Films, hydrogels, and hydrocolloids are some examples, and they present a few advantages such as the capacity to enable the transmission of gases and maintain a moist environment at the wound site, which improves and accelerates the wound healing process [14]. However, these approaches also have some limitations, the possibility of maceration, and the necessity for periodic replacement, and until now, none of them have been capable of fully restoring the skin’s native structure and functions [15,16]. Such emphasizes the need for the development of an efficient wound dressing that can provide the ideal structural and biochemical mechanisms for promoting efficient skin regeneration. Asymmetric membranes, widely explored for filtration and gas separation, recently captured the attention of researchers for being applied as wound dressings [17,18,19]. The utilization of asymmetric wound dressings aims to reproduce a skin-like layered organization, which consists in a top layer to protect the wound site and mechanical support, and a bottom layer that facilitates cell migration, adhesion, and proliferation, and provides a moist environment [20]. Particularly, the nanofibrous composition of asymmetric electrospun membranes allows them to reproduce the extracellular matrix (ECM) structure, thus providing additional anchoring points for cell adhesion and proliferation [21].

In this review, the most common production techniques and key properties of asymmetric membranes were overviewed, focusing on the development of asymmetric electrospun membranes aimed for skin regeneration. Furthermore, the application of asymmetric electrospun membranes to mediate the delivery of biologically active molecules, i.e., antibacterial agents or wound healing enhancers, was also highlighted.

## 2. Asymmetric Membranes

Asymmetric membranes are 3D matrices composed of two layers, which enable a high similarity with the native skin while protecting the wound against bacterial penetration, dehydration, and exudate accumulation. These membranes have been emerging as ideal wound dressings due to their inherent capacity to mimic both epidermal and dermal skin layers [22,23]. In the literature, it is often reported that an ideal wound dressing must act as a protecting barrier, which avoids microorganisms’ invasion as well as additional damages resulting from external hazard agents [3,24,25]. Further, these dressings should also promote cell proliferation and migration, angiogenesis, sustain a moist environment at the wound site, and be compatible with gaseous’ and fluids’ exchanges [25]. Additionally, the wound dressing must be produced with biocompatible and biodegradable materials, be cost-effective, and its production scalable [3]. In this regard, the structural features of the asymmetric membranes fulfil the requirements for their application as dressings in the healing process (as represented in Figure 1). The external (i.e., top) layer of the asymmetric membranes often presents a dense matrix with low total porosity, small pores (i.e., inferior to the bacteria’s size), and a hydrophobic character [26]. Such characteristics provide to the wound site both protection against external agents (e.g., bacteria, radiation) and mechanical stability without compromising the gaseous exchanges, likewise the epidermis layer [27]. In turn, the internal (i.e., bottom) layer is composed of hydrophilic materials and a loose structure with high porosity and large pores [28]. Therefore, this layer facilitates cell migration, adhesion, and proliferation, the absorption of wound exudate, as well as providing a moist environment compatible with nutrient exchange [20].

So far, the production of asymmetric membranes has been accomplished by wet or dry/wet phase inversion, scCO_2_-assisted phase inversion, and electrospinning methodologies [29].

In fact, the wet-phase inversion method was the first technique used to produce membranes with an asymmetric structure [30]. This technique takes advantage of the polymer precipitation in a non-solvent coagulant bath to originate a membrane comprising a compact top layer and a porous sub-layer [31]. Marcano and colleagues reported the development of poly(hydroxyalkanoate)/poly(vinyl pyrrolidone) (PVP) asymmetric membranes using ultrapure water (0.05501 μS × cm^−1^ and 18.18 MΩ × cm) as a coagulation bath [32]. For this purpose, these authors dissolved both polymers in *N*-methyl-2-pyrrolidone, casted onto a glass plate in a 140 µm-thick layer and submersed it in the aqueous coagulation bath for 1 h at room temperature. Furthermore, Marcano and colleagues also reported that increasing concentrations of PVP (porogen polymer), from 0 to 10 and 30%, led to asymmetric membranes with a greater surface pore number, pore size, and pore density. A higher porosity also resulted in an enhanced capacity to encapsulate a protein-based therapeutic (Dispersin B), reaching the maximum of ≈12 µg × cm^−2^ at 30% of PVP. Additionally, these authors also demonstrated that the polyhydroxyalkanoate/PVP (30%) asymmetric membrane inhibited the formation of biofilm by *Staphylococcus epidermidis* (*S. epidermis*) in 33% and mediated the detachment of 26% of the formed biofilm [32]. 

Nevertheless, the asymmetric membranes produced through the wet-phase inversion method present some limitations that hinder their efficacy in promoting the wound healing process, such as a defective and thin top layer that fails in preventing excessive water evaporation and protection against external agents. To address these issues, an evaporation step was added before the polymer immersion in the coagulant bath, originating the dry/wet-phase inversion method [33]. This approach requires the utilization of a volatile solvent that is partially evaporated, which increases the polymer concentration at the top of the film resulting in a denser and less porous top layer [31]. Mi et al. prepared a chitosan (CS) asymmetric membrane enriched with silver sulfadiazine through the dry/wet-phase inversion technique [34]. In this approach, CS was dissolved in an aqueous acetic acid solution (1 wt.%), cast into a mold, incubated at 50 °C for 10 to 60 min, and immersed in the coagulant solution (NaOH (2 wt.%)–Na_2_CO_3_ (0.05 wt.%)) for 24 h. The authors observed an increase in the thickness of the dense top layer and an overall decrease in the porosity of the asymmetric membranes when longer evaporation times were performed. Moreover, such structural alterations resulted in a decrease in the water uptake capacity (900 to 132%) and in the water vapor transmission rate (2800 to 2110 g × m^−2^ × day^−1^) [34]. 

Alternatively, the scCO_2_-assisted phase inversion technique, as a green technology, has also been employed for the production of asymmetric membranes [29]. In this approach, the membrane assembly occurs via the precipitation of a polymeric solution in CO_2_ supercritical conditions [35]. Therefore, the asymmetric membrane properties can be optimized by adjusting the concentration of the casting solution, the ratio of non-solvent/solvent, the pressure, the temperature, and the depressurization rate. As the main advantage, this technique allows the replacement of conventionally used organic solvents by scCO_2_, which is environmentally friendly and can enhance the biocompatibility of the produced wound dressings [36]. Morgado and co-workers developed a poly(vinyl alcohol) (PVA)/CS asymmetric membrane loaded with ibuprofen via the scCO_2_-assisted phase inversion technique, aiming to apply it as a wound dressing [37]. With that in mind, the authors prepared a PVA/CS solution (17.25 wt.%) containing β-cyclodextrins loaded with ibuprofen. Then, the solution was cast into a stainless-steel cap and performed the supercritical immersion precipitation technique (90% CO_2_ and 10% ethanol, flow of 5 mL × min^−1^ for 120 min, 20 MPa, and 45 °C) testing two different depressurization rates (4 and 10 min). The authors reported that both depressurization rates resulted in membranes with similar overall porosity (average pore diameter ≈0.7 µm and porosity 37%), being obtained smaller and more homogeneous pores at 10 min of depressurization. Further, the increase in the depressurization rate from 4 to 10 min also resulted in asymmetric membranes with a denser top layer. Moreover, the asymmetric membranes PVA/CS+β-cyclodextrins loaded with ibuprofen (10 min of depressurization) presented a swelling of 200% and 310% at pH 8 and 5, respectively, and a water vapor transmission rate of 410 g × m^−2^ × day^−1^. In the in vivo studies, the group treated with the asymmetric membranes PVA/CS+β-cyclodextrins loaded with ibuprofen (10 min of depressurization) presented, after 21 days, a higher wound closure than the control, i.e., the wounded area was 2-times inferior and minimized both the scab formation and inflammatory response [37]. However, the development of asymmetric membranes by the scCO_2_-assisted phase inversion technique requires the use of specialized, robust, and costly high-pressure apparatus for achieving the supercritical conditions, which hinders the scalability of the production process, and the translation to the clinic [29].

### 2.1. Electrospinning Technique

The electrospinning apparatus usually used to produce nanofiber-based structures comprises four different components: A syringe pump, a capillary needle, a high voltage power supply, and a metal collector (Figure 2) [38]. During the electrospinning process, an electric field is applied between the needle and the collector [39]. At a critical voltage, the polymeric solution is ejected from the tip of the Taylor cone towards the metal collector. Such promotes the solvent evaporation and the deposition of polymeric fibers in the collector [40]. The properties of the produced polymeric structures are controlled by the feeding solution (e.g., concentration, viscosity, surface tension, solvent volatility, and conductivity), environmental conditions (e.g., temperature and humidity), and operating parameters (e.g., voltage, solution flow rate, and needle-to-collector distance) [41]. Such will directly impact on the fibers’ diameter (usually at nanoscale) and arrangement, mechanical strength, and structural porosity [42,43]. Further, the possibility to select different types of tips (e.g., co-axial, multi-jet, and multifluidic co-axial) and collectors (e.g., plain or grid-like structure as well as stationary or rotating) influences the fibers’ topography and spatial arrangement [39]. Moreover, the electrospinning technique can also be optimized to be compatible with the cell encapsulation enhancing the wound healing capacity, contrasting with the other techniques explored for producing asymmetric polymeric membranes [44,45,46,47]. For example, Alfare De Prá et al. observed that the deposition of poly(caprolactone) (PCL) fibers in a stationary cylinder resulted in randomly oriented fibers with an average diameter of 1142 ± 391 nm [48]. Otherwise, when the electrospinning process was performed with the cylinder collector at a rotation speed of 2000 rpm, the mean fiber diameter decreased to 663 ± 334 nm and presented a more homogeneous size distribution and fiber orientation. In wound healing applications, the fibers present in the electrospun membranes are aimed to mimic the interconnected 3D network of extracellular matrix of native skin [49]. Additionally, the high surface area-to-volume ratio characteristic of electrospun membranes facilitates the cell adhesion and proliferation [50].

The electrospinning technique presents the advantage of being compatible with natural, semi-synthetic, or synthetic polymers, which facilitates the development of multifunctional dressings [51]. In this way, the polymers’ selection for wound healing applications must be based on their biocompatibility, biodegradability, hydrophilicity, and mechanical properties [51,52].

The natural polymers are recognized by their superior biocompatibility and possible interaction with cells via cell surface adhesion receptors which enable the support of cell adhesion and proliferation [24]. One reason for this feature is its similarity with some macromolecules found in the human body or its presence in the ECM [51]. CS [26], hyaluronic acid (HA) [28], collagen [53], gelatin (Gel) [54], and silk [20] are some examples of natural polymers usually employed to produce nanofibrous mats. However, natural polymers are usually associated with sub-optimal mechanical strength, faster degradation profiles, and higher costs [52]. On the other hand, synthetic polymers are widely used due to their good mechanical properties (like elasticity and stiffness) and tunable biodegradability. Moreover, the synthesis of these polymers can be scaled up, lowering the costs [52]. Some examples of synthetic polymers used in the electrospinning technique are PCL [55], PVA [49], poly(lactide-*co*-glycolide) (PLGA) [56], and poly(l,d-lactic acid) (PLA) [57]. Otherwise, these polymers also have some limitations, such as their lower biocompatibility as well as the lack of cell-specific recognition and attachment moieties [52].

In this way, researchers often adopt the production of hybrid nanofibrous mats, i.e., blends of synthetic and natural polymers, which allow the best compromise between the physical and biological properties to be found [58]. Furthermore, in the case of the asymmetric electrospun membranes, the selection of the polymers must take into consideration the different properties desired for the bottom and top layers. The external (top) layer should present a hydrophobic character (use of hydrophobic polymers) to avoid bacterial contamination and displays excellent mechanical properties (use of synthetic polymers) [22]. Otherwise, some studies demonstrated that the cells more easily adhere and proliferate in moderate hydrophilic wound dressings than in hydrophilic or hydrophobic ones [59]. Therefore, the inner (bottom) layer usually displays a hydrophilic character facilitating the cell adhesion, proliferation, and migration [60]. Such also enables the capacity to absorb the wound exudate, and consequently maintain the adequate moist environment in the wound site [61].

### 2.2. Asymmetric Electrospun Membranes

The production of asymmetric electrospun membranes can also be accomplished by promoting the layer-by-layer assembly of different nanofiber-based structures (Table 1) [62]. Such is usually accomplished through the formation of one layer on the top of an already produced layer, hence mimicking both epidermis and dermis layers of the skin [62]. The external layer is often composed of smaller-sized hydrophobic nanofibers, which allow for the construction of a denser structure with small pores that protects the wound site against microorganisms invasion [27]. Additionally, polymers with strong mechanical properties, such as PCL and poly(l-lactic acid), are usually selected for assembling this layer since these materials confer mechanical resistance to the asymmetric electrospun membrane [55]. Otherwise, the bottom layer presents a porous interconnected 3D network with hydrophilic nanofibers that mimics the ECM [54]. Such promotes a moist environment that supports cell migration, adhesion, and proliferation [43,63]. Therefore, the combination of these two electrospun layers results in asymmetric membranes with enhanced biological performance (Table 2) [21,42,43]. 

Chanda et al. produced a CS-PCL/HA asymmetric electrospun construct to obtain an asymmetrical membrane with enhanced mechanical stability and capacity to perform wound bed hydration [26]. In this process, the HA-poly(ethylene oxide) solution was electrospun in the assembled CS-PCL nanofibrous top layers and crosslinked with glutaraldehyde vapor to obtain the bilayered (CS-PCL/HA) membrane. The CS-PCL/HA asymmetric membrane exhibited fibers with an average diameter of 362.2 ± 236 nm, which are similar to those found in collagen fibers of ECM (50 to 500 nm). The membranes presented an overall porosity superior to 90% and a water vapor transmission rate of ≈2500 g × m^−2^ × day^−1^. Further, the authors reported that the water contact angle decreased from 127 ± 2° in single PCL electrospun membrane to 82.4 ± 6.4° in the CS-PCL/HA asymmetric membrane, a feature that is responsible for an increase in the swelling capacity from 105 to 135% (Figure 3). Otherwise, the CS-PCL/HA asymmetric membranes presented a bacterial adhesion 3.9-times inferior to the single PCL electrospun membrane and increased cytocompatibility [26].

Similarly, Chen et al. developed a CS/poly(l,d-lactic acid) (PDLLA) asymmetric electrospun membrane [64]. For this purpose, PDLLA nanofibers were randomly deposited in a static collector creating the bottom layer, which was subsequently coated with aligned CS nanofibers using a rotating collector at 1000 rpm. The resulting asymmetric membrane presented a CS top layer with small pore size and dense structure due to the packed small-sized nanofibers (i.e., 243 nm) covering a highly porous PDLLA fibrous layer (2.8 μm average fiber diameter and a surface pore size of about 12.18 ± 1.7 μm). These authors observed that the CS/PDLLA asymmetric electrospun membrane presented enhanced HT1080 cell viability and infiltration when compared to single PDLLA fibrous mats and films. Moreover, in in vivo assays, the CS/PDLLA dual-layer membrane allowed the cell infiltration up to 32.5 μm in depth, which contributed to the restoration of both epidermis and dermis layers by enhancing the regeneration of keratinocytes and fibroblasts, respectively. In this way, this asymmetric membrane allowed a faster restoration of the skin’s structure and function when compared to the gauze, PDLLA film, or fibers [64]. 

In a different approach, Wu et al. developed an asymmetric electrospun membrane composed of a hydrophobic top layer with β-glucan butyrate nanofibers and a hydrophilic β-glucan acetate (BGA) nanofibrous inner layer (BGE-B asymmetric membrane) [61]. The assembly of the BGE-B membrane was achieved through the electrospinning of a β-glucan butyrate (BGB) solution onto the already assembled BGA nanofibrous mats. The BGA nanofibrous mats presented an average width of 150 ± 58 nm, a hydrophilic character (contact angle decreased from 86 to 0° within 10 s), and a high swelling capacity (400%). In turn, the BGB layer presented a mean fiber diameter of 410 ± 186 nm, high hydrophobicity (contact angle of 126.4°), and low swelling ratio (≈20%). The authors also reported that NIH 3T3 fibroblasts could adhere and proliferate on the BGE-B membrane, mainly in the BGA layer, whereas HaCaTs keratinocytes showed a homogeneous proliferation in both sides of the asymmetric membrane. Moreover, the in vivo assays performed on a full-thickness mouse skin wound model demonstrated that on day 14, the group treated with BGE-B membrane exhibited a reduction in the wounded area of 83.1%, contrasting with the 57.5 and 26.2% obtained in the gauze and control groups, respectively. Further, histological data revealed that the BGE-B membrane increased the epithelization process, and the structure of the newly formed tissue was quite similar to that of normal skin [61].

## 3. Electrospun Asymmetric Membranes as Delivery Systems of Biomolecules

Apart from the structural advantages of asymmetric electrospun membranes, the nanofibers present in their structure can also incorporate bioactive agents for increasing its antibacterial efficacy and/or enhance the wound healing process (Figure 4). Table 3 provides an overview of the biologic properties of asymmetric membranes produced through electrospinning aimed to be used as wound dressings.

The electrospun membranes are recognized as good drug carriers due to their high surface area to volume ratio, connected porosity, high drug loading capacity, and tunable release profile [72]. The release of biomolecules from the nanofibers occurs mainly through three mechanisms desorption from the surface, diffusion through fibers, and fiber degradation. The desorption consists of a burst release due to the proximity between the biomolecule encapsulated on the fiber surface and the surrounding liquid [52]. The mechanism based on the biomolecules’ diffusion is characterized by their transport through the channels and pores of the fibers, mainly in the case of non-biodegradable polymers [52,73]. Otherwise, the use of biodegradable polymers leads to the release of the biomolecule due to the degradation of the polymeric matrix. This degradation creates additional spaces in the nanofibers’ structure, which facilitates the release of the encapsulated biomolecules into the surrounding medium [52,73]. Additionally, the drug release profile can also be affected by several factors, such as the fibers’ composition (hydrophilicity, hydrophobicity, and biodegradability of the polymers and biomolecules), the interactions between the polymer/biomolecule/solvent, and the incorporation technique [74,75]. Furthermore, the fiber structure, i.e., morphology, diameter, porosity, and ratio (polymer-encapsulated drug), can also affect the release [72]. In respect to the composition, hydrophilic drugs present a more effective encapsulation and homogeneous distribution within hydrophilic polymers, while the utilization of hydrophobic polymers shows better results to encapsulate hydrophobic drugs [52,76]. It is worth noticing that hydrophilic drugs usually present faster drug releases (i.e., burst release) due to their high solubility in the release media [52]. Furthermore, changes in the matrix structure can also influence drug release. For example, an increase in the surface area-to-volume ration and porosity of fibrous mats leads to a burst release of the drugs from the nanofibers [77]. Moreover, the development of fibers with a multilayer structure enables a more sustained release, or even the encapsulation of different therapeutic agents in different layers, each one presenting a release profile according to its location in the fiber [62]. Another strategy that has been emerging is the use of stimuli-responsive materials which facilitate the control of the drug release both temporally and spatially [51,78]. Temperature [79], ultrasound [80], light [81], and endogenous changes in the pH value [82], are some examples of the stimulus that can be explored to trigger the drug release by inducing changes in the polymeric nanofibers.

The incorporation of therapeutic or antimicrobial agents can be performed before (e.g., blend, co-axial, and emulsion) or after (e.g., physical adsorption, layer-by-layer assembly, and chemical immobilization) the electrospinning process [43]. In the blend electrospinning, the drug is incorporated in the polymeric solution and the nanofibers are produced with the drug uniformly distributed within their structure [83]. The resulting nanofibrous mats often exhibit a burst release dependent on polymer degradation and drug diffusion from the nanofiber [24]. Alternatively, core-shell nanofibers encapsulate the therapeutic agents within the nanofiber core, which is enclosed in an external polymeric layer that provides a more controlled and prolonged drug release [51]. These nanostructures can be produced by co-axial or by emulsion electrospinning. The co-axial electrospinning uses concentric needles to create a layer-by-layer organization, while the emulsion relies on the utilization of an aqueous solution and an oil phase that are emulsified together to create the core-shell nanofibers [75,76]. Otherwise, the drug loading can be achieved via post-electrospinning techniques that comprise the adsorption of the drug onto the surface of the fibers through non-covalent or covalent interactions [76]. The physical adsorption of therapeutic agents is based on the creation of non-covalent interactions (electrostatic and/or hydrophobic) between the nanofibers and the drug, while the chemical approaches require the immobilization of the therapeutic agents via covalent bonds. Such enables the development of nanofiber membranes with different drug release profiles, i.e., usually non-covalent interactions lead to a faster and poorer control over the drug release, whereas a sustained/stimuli-responsive release can be achieved using covalent binding [24,76]. These different approaches allow the selection of the optimal loading method according to the bioactive agent as well as the possibility to develop wound dressings with tailored release profiles [24]. For example, Buck et al. compared the antimicrobial and release properties of PLGA electrospun fibers incorporating ciprofloxacin (CIP) through blend or physical adsorption [56]. The blend nanofibers were produced by the electrospinning of a CIP-PLGA solution, whereas in the physical adsorption PLGA nanofibrous mats were immersed in a CIP solution and dried. The authors observed that upon the immersion of PLGA mats in phosphate-buffered saline solution (PBS) (pH 7.4), at 37 °C, the CIP loading by physical absorption resulted in a fast release, reaching its maximum at 6 h. In turn, the blend counterparts presented a sustained release for 48 h. Moreover, the authors also reported that the antibacterial capacity against *Pseudomonas aeruginosa* (*P. aeruginosa*), *Staphylococcus aureus* (*S. aureus*), and *S. epidermidis* was influenced by the loading method. In fact, the physical absorption resulted in higher-sized inhibitory halos, whereas the blended disks retained a higher percentage of the inhibitory zone after 48 h of incubation [56]. In turn, Jin and co-workers compared the incorporation of multiple epidermal induction factors (EIF) by coaxial electrospinning and blend electrospinning, in poly(l-lactic acid)-*co*-poly(ε-caprolactone) (PLLCL) and Gel nanofibers (Gel-PLLCL-EIF (coaxial electrospinning) and Gel-PLLCL-EIF (blend electrospinning), respectively) [84]. To accomplish that, core-shell nanofibers were produced by electrospinning a Gel-PLLCL mixture as an outer solution and 5% bovine serum albumin with a concentrated epidermal induction medium (CEIM, composed of epidermal growth factor (EGF), insulin, hydrocortisone, and retinoic acid) as core solution. Alternatively, the Gel-PLLCL-EIF (blend electrospinning) nanofibers were produced through the conventional electrospinning of the Gel-PLLCL-CEIM blend. These authors observed that the Gel-PLLCL-EIF (blend electrospinning) released 44.9% of its content in the first 3 days, reaching the maximum of 77.8% at day 15, whereas the Gel-PLLCL-EIF (coaxial electrospinning) presented a stable and sustained release with 50.9% of EGF released at day 15. Such differences were attributed to the Gel-PLLCL outer layer in the core-shell nanofibers that acted as a barrier that diminished the initial burst release [84]. Apart from the structural organization, the drug release of the nanofibrous mats could also be controlled by the biodegradation profile of the polymers or using smart materials responsive to different stimuli [85]. For more information regarding the utilization of nanofibrous structures as drug delivery systems the readers are referred to [72,85,86,87].

### 3.1. Electrospun Asymmetric Membranes with Antibacterial Activity

When the skin integrity is disrupted, the occurrence of infections leads to the deterioration of the granulation tissue, growth factor, and ECM components and, consequently, to the impairment of the wound healing process [85]. These infections can be caused by different bacteria, usually in initial stages the infections occur as a consequence of Gram-positive bacteria, such as *S. aureus* and *Streptococcus pyogenes* (*S. pyogenes*), while in later stages they are originated by Gram-negative bacteria, such as *Escherichia coli* (*E. coli*) and *P. aeruginosa*. In this way, the incorporation of antibiotics (e.g., CIP, gentamicin, and sulfadiazine), nanoparticles (e.g., silver, iron oxide, nitric oxide), and natural products (e.g., honey, essential oils, and CS) in wound dressing has been widely explored to prevent bacterial infections [14,85]. 

Among the different antibacterial agents, the incorporation of antibiotics in asymmetric electrospun membranes has been one of the most explored approaches [69,71]. Zhao et al. produced an asymmetric poly(l-lactide) (PLLA)-sericin (SS)/PLLA electrospun membrane loaded with nitrofurazone (NFZ) for wound dressing applications [71]. In this process, the NFZ was blended in the PLLA-SS and PLLA solutions before the electrospinning process. The NFZ-loaded PLLA bottom layer was produced over the NFZ-loaded PLLA-SS nanofibrous mats. The PLLA bottom layer presented fibers with an average diameter of 814 nm. In turn, in the PLLA-SS mats, the average diameter of the nanofibers increased from 413 to 1095 nm by changing the PLLA-SS ratio from 4:1 to 1:1. Despite this variation, the asymmetric membranes presented a similar overall porosity ranging from 75.14 ± 5.43% to 78.35 ± 2.38%. The authors observed that the NFZ presented a release profile dependent on the nanofibrous layer. The top PLLA-SS nanofibrous mats presented a fast release profile with more than 98% of NFZ detected in 10 min of incubation, independently of the PLLA-SS ratio. Otherwise, the PLLA bottom layer presented a more controlled and sustained release, reaching the 17.6% after 48 h. Such difference is attributed to the possible interaction of NFZ with SS, a good water-soluble material that allows a faster drug diffusion upon its dissolution in the media. Moreover, the initial burst release of NFZ from the PLLA-SS layer is important for the elimination of bacteria that can be initially present on the wound site, whereas the more sustained release from the PLLA bottom layer can contribute to long-term antibacterial effects. The studies performed on *E. coli* and *Bacillus subtilis* demonstrated the SS intrinsic antibacterial activity, which was enhanced with the NFZ incorporation (larger inhibitory halos). In in vivo studies, the group treated with the NFZ-loaded asymmetric membranes showed a faster wound healing, i.e., 97 and 84% wound size reduction for the NFZ-loaded dual-layer membranes and commercial woven dressing, respectively [71].

Nevertheless, the rise of multidrug-resistant bacteria over the past years has highlighted the necessity to select and study new antibacterial agents. In this way, natural products have been screened to identify alternative antibacterial approaches. Among them, essential oils have been recognized due to their antioxidant, antiviral, anticancer, insecticidal, anti-inflammatory, anti-allergic, and antimicrobial properties [85]. The antibacterial activity of these natural products is mainly attributed to the phenolic compounds, specifically to thymol (THY) and carvacrol (CRV) [20,66]. Miguel et al. produced a silk fibroin (SF)-based asymmetric electrospun membrane loaded with THY for being applied in the wound healing [20]. The THY was blended in a SF-HA solution that was electrospun over the SF-PCL top layer and treated with ethanol vapor to improve the water stability of the SF. The SF-based asymmetric electrospun membrane presented an overall porosity of 74.78 ± 6.98%, a swelling capacity of 400%, and nanofibers with a mean diameter of 615.9 ± 190.4 and 412.7 ± 106.7 nm in the top SF-PCL and bottom SF-HA-THY layers, respectively. Moreover, the authors observed a burst release of THY in the first 8 h, reaching a maximum of 91.87 ± 0.99% at pH 8 (Figure 5). Otherwise, the antibacterial assays performed with and *P. aeruginosa* revealed that the SF-PCL top membrane can avoid the infiltration of both bacteria through the bottom layer with an efficacy almost similar to the conventional filter paper. Further, the SF-HA-THY bottom membrane significantly inhibited the proliferation of *S. aureus* and *P. aeruginosa* (bacterial growth inhibition of 87.42% and 58.43%, respectively) when compared to the SF-HA membrane (4.05% and 3.42%) [20]. 

In a similar approach, Aragón and co-workers produced a PCL/poly(vinyl acetate) (PVAc) asymmetric membrane loaded with CRV for increasing the antimicrobial capacity of the wound dressing [66]. The production of the asymmetric membrane was accomplished in a three-step process using a multi-jet electrospinning apparatus: 1) Deposition of PCL top layer; 2) multi-jet deposition of PCL and PVAc-CRV blend; and 3) deposition of PVAc-CRV bottom layer. The PCL/PVAc asymmetric membrane presented an overall porosity superior to 80%, comprising a denser hydrophobic PCL top layer with a thickness around 100–120 μm and fibers with an average diameter of 360 ± 68 nm, a transition layer, and a loose spiderweb PVAc-CRV nanofibrous structure with an average diameter of 600 ± 100 nm. Moreover, the CRV release assays performed in conditions mimicking the wound site (pH 8 to 5) showed an initial burst release ≈45% after 24 h, at pH 8, followed by a sustained diffusion up to 85% at day 21 and pH progression from 8 to 7.4 and 5. Moreover, the authors reported that the PCL/PVAc-CRV asymmetric membrane significantly inhibited the proliferation of *E. coli* (from 1.6 × 10^9^ to 1.2 × 10^7^) and *S. aureus* (5.7 × 10^10^ to 2.3 × 10^7^) [66].

### 3.2. Electrospun Asymmetric Membranes Loaded with Bioactive Molecules that Improve the Healing Process

The wound healing process involves five different phases, namely hemostasis, inflammation, migration, proliferation, and remodeling. This complex process is based on a complex interaction between cells, growth factors, and cytokines [88]. In this way, tissue engineering researchers have been incorporating bioactive molecules in wound dressings to promote and improve the different phases of the healing process. 

The delivery of vitamins can stimulate cell migration to the wound site, increase the collagen synthesis, enhance the angiogenesis, and modulate the inflammatory response leading to an improved skin regeneration [89]. Zahid and collaborators produced a PCL/PLA electrospun membrane enriched with α-tocopherol acetate (vitamin E derivative-VE) for supporting the wound healing process [57]. The VE was blended both with PCL and PLA solutions, then the PLA-VE blend was electrospun over the assembled PCL-VE top layer using a rotating collector. The asymmetric membrane showed a sustained release of VE over 21 days reaching a maximum of 78%. Further, the authors observed that the VE-loaded PCL/PLA asymmetric membrane was capable of promoting the NIH3T3 fibroblasts cell migration, adhesion, and proliferation. Additionally, the studies performed on chick chorioallantoic membrane showed that the group treated with VE-loaded PCL/PLA asymmetric membrane presented a higher number of blood vessels, i.e., 2-times superior to that detected on non-loaded PCL/PLA bilayers (Figure 6).

Otherwise, the asymmetric membranes can also be loaded with anti-inflammatory agents, such as curcumin, chrysin, ibuprofen, and pioglitazone (Pio) [14,54,90]. A chronic or uncontrolled acute inflammatory process could result in excessive production of inflammatory mediators, free radicals, and cytotoxic enzymes that could negatively impact on the healing process [85]. Yu et al. developed a poly(ε-caprolactone) (PeCL)/Gel nanofibrous asymmetric membrane loaded with Pio using a 40 µm-pore-size nylon mesh template [54]. The hydrophobic PeCL layer was electrospun on the top side of the nylon mesh template, whereas the hydrophilic Gel-Pio blend was produced on the bottom side and crosslinked with genipin. The resulting membrane is composed of 270 nm diameter PeCL nanofibers on top and 144 nm Gel nanofibers in the bottom layer. The Pio release from the PeCL/Gel asymmetric membrane rapidly reaches 40% in day 1 and stabilizes around 75% until day 14. Bacterial adhesion tests performed with *S. aureus*, *P. aeruginosa*, and *E. coli* demonstrated that the PeCL top layer significantly reduced the number of adhered bacteria, particularly for *P. aeruginosa* and *E. coli* strains, due to its hydrophobicity (surface water contact angle of 145°). Furthermore, when compared to the non-loaded counterparts, the PeCL/Gel-Pio asymmetric membranes promoted the migration of both human skin fibroblasts and human umbilical vein endothelial cells. Moreover, the authors also observed a clear reticular vascular structure with more tube junctions and nodes in the HUVECs group treated with PeCL/Gel-Pio demonstrating its proangiogenic capacity. The in vivo assays performed on type 2 diabetic mice showed that the wounds treated with the PeCL/Gel-Pio asymmetric membrane were almost fully closed on day 10, whereas the same was only observed at day 14 in the groups treated with PeCL/Gel and Tegaderm. Further, at day 14, the PeCL/Gel-Pio group presented a relative collagen content almost 2-times superior to the remaining groups, being also observed in histological images’ regular and orderly collagen arrangement as well as a smaller granulation tissue space [54].

## 4. Conclusions and Future Perspectives

Acute and chronic skin injuries continue to be a major health issue for the worldwide population. In this field, the electrospun asymmetric membranes have been emerging in recent years as promising wound dressings. These wound dressings are usually comprised by two different layers, (i) an epidermis-like layer with a dense structure that confers protection and mechanical stability, and (ii) a dermis-like layer with a loosened and porous structure that provides support for cell migration and proliferation, stimulates the angiogenesis, and allows the gaseous and fluid exchanges. Further, the nanofibrous structure of electrospun asymmetric membranes highly resembles the interconnected 3D network of the ECM and the native physical fibrillar structure of collagen. Additionally, the loading of bioactive molecules on electrospun asymmetric membranes can be achieved through different approaches (pre- or post- the electrospinning process). 

Despite the electrospun asymmetric membranes’ capacity to actively support, stimulate, and accelerate the wound healing process, further developments are still required to achieve optimal skin regeneration in humans. Although there are no asymmetric electrospun membranes in clinical trials, asymmetric structures (e.g., Apligraf^®^, PolyActive^®^, PermaDerm^™^, and OrCel^®^) have been used already as dermal/epidermal wound substitutes, demonstrating the potential of asymmetric membranes for application in skin regeneration, namely in the healing of full-thickness wounds. Furthermore, the incorporation of growth factors and proteins are already under development in electrospun membranes; however, the inclusion of them in asymmetric electrospun membranes may be an important factor to enhance its biological performance. In addition, the electrospun asymmetric membranes must be designed in a way that the release of the bioactive agents occurs accordingly to the wound healing stage that they will assist. Further, in the future, the incorporation of stem cells will be determinant to obtain a fully recovered skin when extensive lesions occur, e.g., regeneration of skin appendages such as hair follicles and shafts, which beyond the aesthetical role are also pivotal for the skin permeability and pigmentation. Moreover, the incorporation of multifunctional and smart materials may allow the real-time monitoring of the wound site environment (e.g., pH, redox state, inflammatory process, and infections), thus facilitating the identification and response to possible complications during the healing process. Finally, the inclusion of electrical stimulation, mechanical stress, or pulsed magnetic field may also improve the wound healing process.

## Figures and Tables

**Figure 1 pharmaceutics-13-00183-f001:**
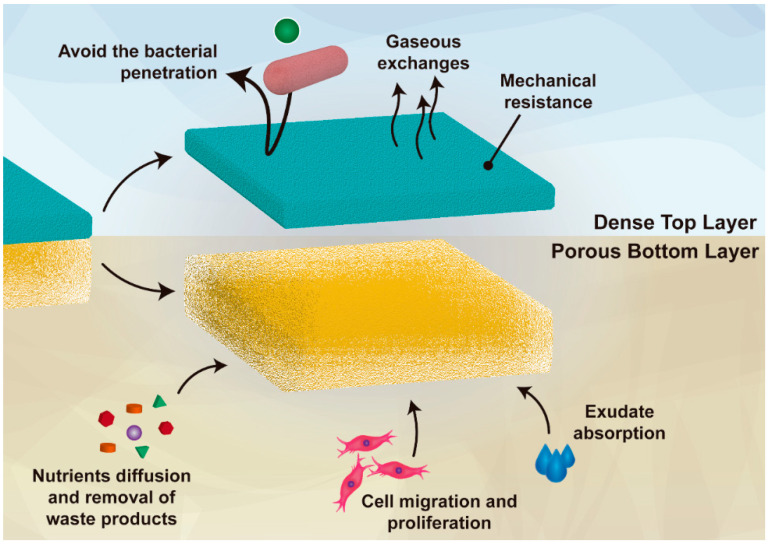
Illustration of the main functions of both layers (top and bottom) of an asymmetric membrane.

**Figure 2 pharmaceutics-13-00183-f002:**
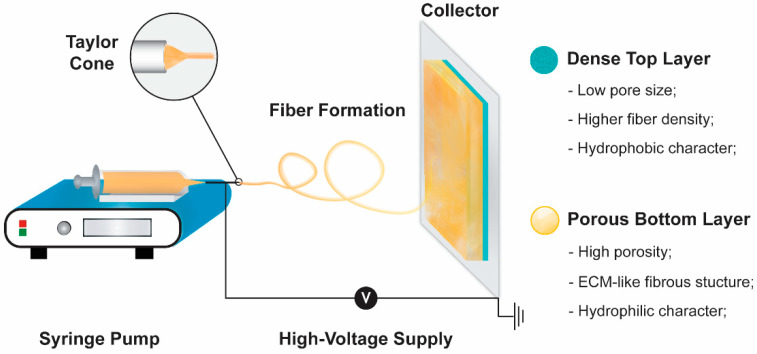
Illustration of a conventional electrospinning apparatus used in the production of an asymmetric electrospun membrane. The physical properties of each layer of the produced membrane are also presented.

**Figure 3 pharmaceutics-13-00183-f003:**
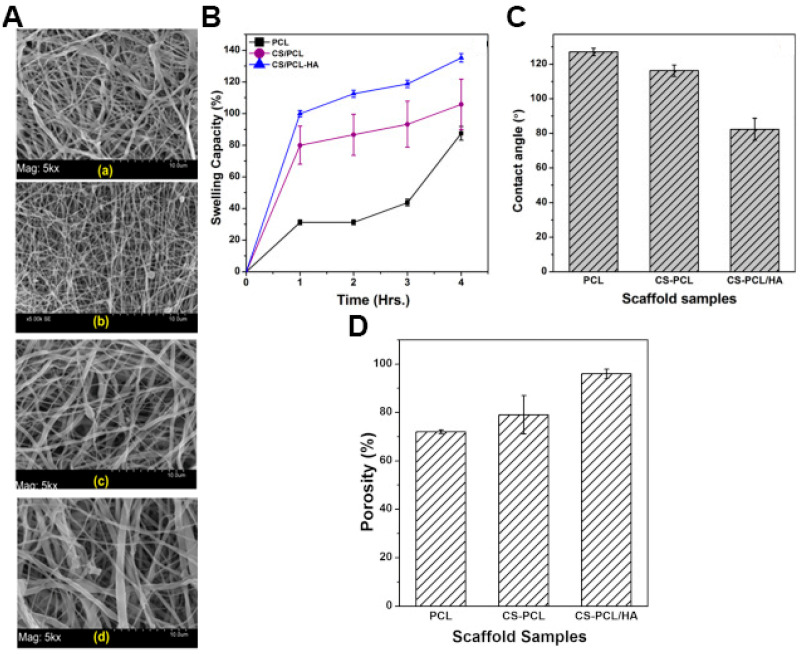
(**A**) SEM photomicrographs, with magnification of 5000×, of electrospun scaffolds of (a) poly(caprolactone) (PCL), (b) chitosan (CS)/PCL, (c) hyaluronic acid/polyethylene oxide (HA/PEO), (d) CS/PCL-HA. Physicochemical properties of only PCL membrane, CS-PCL membrane (top layer), and CS-PCL/HA asymmetric membrane; (**B**) swelling capacity of the scaffolds; (**C**) water contact angles of the scaffolds; (**D**) porosity of the scaffolds. Reprinted with permission from [26], Elsevier, 2018.

**Figure 4 pharmaceutics-13-00183-f004:**
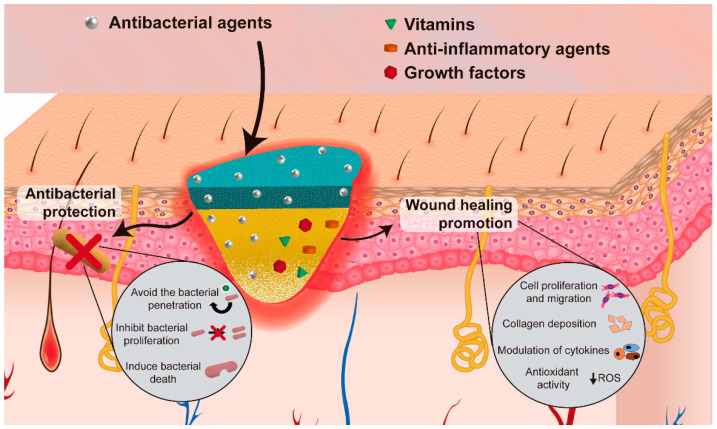
Illustration of the incorporation of bioactive agents to improve the antibacterial efficacy and/or enhance the wound healing process.

**Figure 5 pharmaceutics-13-00183-f005:**
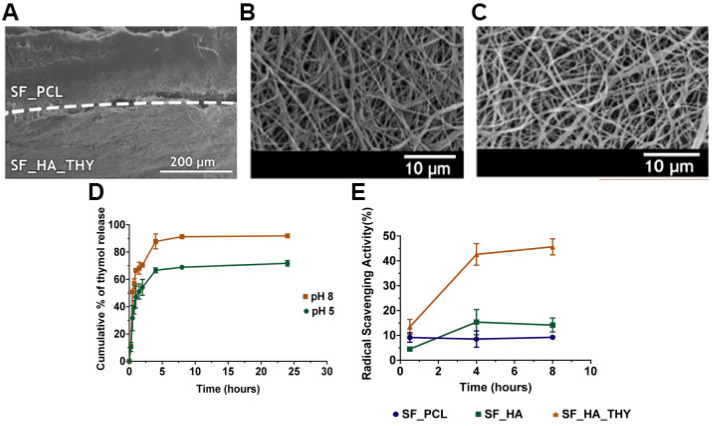
SEM images of (**A**) produced asymmetric membrane cross-section (top layer: SF_PCL; bottom layer: SF_HA_THY); (**B**) SF_PCL after the crosslinking process; (**C**) SF_HA_THY after the crosslinking process; (**D**) evaluation of the THY In Vitro release profile; (**E**) evaluation of the antioxidant activity of the SF-PCL membrane (top layer), SF-HA membrane (bottom layer) with or without THY. Reprinted with permission from [20], Elsevier, 2019.

**Figure 6 pharmaceutics-13-00183-f006:**
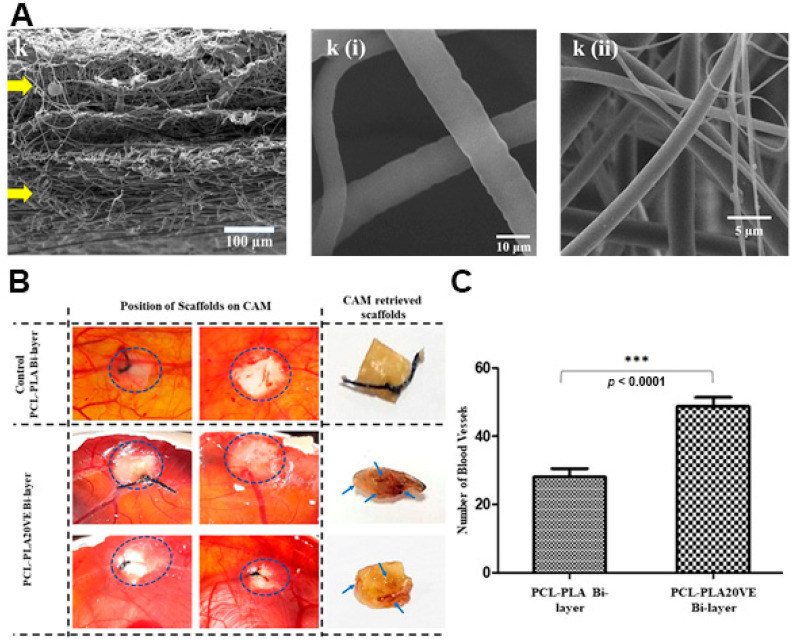
(**A**) SEM images of cross section of PCL-PLA20VE Bi-layer; k(i) PCL20VE; k(ii) PLA20VE. (**B**) Evaluation of angiogenic potential of the PCL/PLA asymmetric membranes with and without 20% of VE. The appearance and position of implanted membranes on CAM, at day 14 of fertilization, indicated by the circles. Blue arrows, on retrieved membranes, depict blood vessels infiltrated inside the explanted membrane. (**C**) Quantification of CAM assay by counting blood vessels around and inside the membranes, from the images taken at day 14, just before retrieving the membranes and sacrificing the eggs. The results are mean ± S.D. (*** *p* < 0.0001) of 4 viable chicks surviving from original group of 7 fertilized eggs per group. Reprinted with permission from [57], Elsevier, 2019.

**Table 1 pharmaceutics-13-00183-t001:** Asymmetric membranes produced through electrospinning aimed to be used as wound dressings. Note: The merged cells report the values obtained for the asymmetric electrospun membranes.

Main Polymer in Top Layer	Layer	Composition of the Layers	Fibers Average Diameter (nm)	Porosity (%)	Water Contact Angle	WVTR (g × m^−2^ × day^−1^)	Swelling Capacity	Degradation Profile	Ref
PCL	Top layer	PCL; PLA	691 ± 282	N.A.	126.8 ± 5.7°	N.A.	N.A.	1.54 ± 0.22%, PBS, pH 7.4, 3 days	[55]
	Bottom layer	GelMA; ChMA	477 ± 228		88.2 ± 6.1° to 46.7 ± 2.2° after 20 s			5.63 ± 0.21%, PBS, pH 7.4, 3 days	
	Top layer	CS; PCL	370 ± 264	≈96.1	82.4 ± 6.4°	≈ 2536	≈135.2%, PBS, pH 7.4, 4 h	30%, PBS, pH 5.5, 15 days ≈4.8%, PBS, pH 7.4, 15 days	[26]
	Bottom layer	HA; PEO	136 ± 61						
	Top layer	PCL	300 ± 50	N.A.	160°	N.A.	N.A.	N.A.	[65]
	Bottom layer	SAG; ZnO nanoparticles	100 ± 30		30°				
	Top layer	PCL	360 ± 68	N.A.	Sample I: 135° Sample II: 120° to 119° after 20 s	Sample I: 2280 ± 153–2992 ± 72 Sample II: 2506 ± 42–4552 ± 82	Sample I: 400% in TRIS, pH 8, 2 h; after 24 h, medium changed to AES, pH 5, swelling decreased to 250% Sample II: 350%, TRIS, pH 8, 2 h; after 24 h, medium changed to AES, pH 5, swelling decreased to 300%	Sample I: ≈2.9% in TRIS, pH 8; after 1 week, medium changed to PBS, pH 7.4, weight loss increased to 3.5%; after another 1 week, medium changed to AES, pH 5, weight loss increased to ≈4.2% Sample II: ≈4.0% in TRIS, pH 8; after 1 week medium changed to PBS, pH 7.4, weight loss decreased to ≈3.6%; after another 1 week, medium changed to AES, pH 5, weight loss increased to ≈4.6%	[66]
	Bottom layer	PVAc Sample I-PVAc in DMF/ETOH; Sample II-PVAc in DMF)	Sample I: 600 ± 100 Sample II: 3000 ± 1000		Sample: 120° to 111° after 20 s Sample II: 76° to 27° after 20 s				
	Top layer	HA; PCL	472 ± 192	90.40 ± 4.25	120.20 ± 0.85°	1762.91 ± 187.50	N.A.	≈10%, PBS, pH 5.5, 7 days	[28]
	Bottom layer	CS; ZN; SA	530 ± 180		101.96 ± 5.08°				
	Top layer	PCL	385 ± 134	55 ± 5	126.2 ± 1.21°	1252.35 ± 21.22	Ratio ≈20, PBS, pH 5, 30 days	≈30%, PBS containing lysozyme, 30 days	[67]
	Bottom layer	CS; AV; PEO	152 ± 54	97.8 ± 4.5	69.06 ± 3.78°				
	Top layer	SF; PCL	615.9 ± 190.4	74.78 ± 6.98	103.10 ± 6.57°	2070.62 ± 102.52	Ratio ≈42, PBS, pH 8, Ratio ≈39.4, PBS, pH 5	23%, PBS, pH 7.4, 7 days	[20]
	Bottom layer	SF; HA; THY	412.7 ± 106.7		38.77 ± 5.32°				
	Top layer	PCL; mupirocin	1031 ± 227	≈ 78.2	34.6 ± 4.6°	N.A.	622%, PBS, pH 7.4, 24 h	N.A.	[68]
	Bottom layer	CS; LID	735 ± 152						
PeCL	Top layer	PeCL	539	N.A.	117°	N.A.	N.A.	N.A.	[69]
	Bottom layer	PDO; TiO_2_ nanoparticles (concentration of 3% (PP3T5T) and 5% (PP5T5T)); TTC	PP3T5T: 611 PP5T5T: 679		PP3T5T - 48° PP5T5T - 79°				
	Top layer	PeCL (nylon mesh pore size 40 (PeCL40) or 80 μm (PeCL80))	270	N.A.	PeCL40: 145.24 ± 3.10° PeCL80: 138.88 ± 3.64°	1556.66 ± 37.79	N.A.	N.A.	[54]
	Bottom layer	Gel	144.9 ± 56.92		39.93 ± 3.03°				
PLA	Top layer	PLA; VE	2150	N.A.	N.A.	N.A.	≈11.82%, PBS, pH 7–7.6, 24 h	≈5.7% in PBS, ≈8.2%, in lysozyme solution, and ≈1.5% in H_2_O_2_, after 21 days	[57]
	Bottom layer	PCL; VE	6094						
BGB	Top layer	BGB	410 ± 186	N.A.	126.4°	N.A.	≈50%, deionized water, 24 h	N.A.	[61]
	Bottom layer	BGA	150 ± 58		86 to 0° after 10 s		≈400%, deionized water, 24 h		
PVDF	Top layer	PVDF	207	N.A.	N.A.	N.A.	Sample I: ≈34.62%, sample II: ≈41.92%, and sample III: ≈48.85%, in PBS, pH 7.4, 336 h	Sample I: ≈31.53%, sample II: ≈39.29%, and sample III: ≈41.18%, in PBS, pH 7.4, 336 h	[70]
	Bottom layer	PHB; CS Sample I: PHB90%-CS10%; sample II: PHB85%-CS15%; sample III: PHB80%-CS20%	100						
PLLA	Top layer	PLLA; SS (1:1, 2:1, 4:1); NFZ (0.2%, 0.5%, 1.0%)	413 to 1095	75.14 ± 5.43 to 78.35 ± 2.38	≈60.0 to ≈143.3°	3161.45 ± 64.97 to 3289.40 ± 58.11	N.A.	N.A.	[71]
	Bottom layer	NFZ; PLLA	814.36 ± 9.93		125.7°				

AV: Aloe vera; AES: Artificial exudates solutions; BGA: β-glucan acetate; BGB: β-glucan butyrate; ChMA: Methacrylated chitosan; CS: Chitosan; Gel: Gelatin; GelMA: Methacrylated gelatin; HA: Hyaluronic acid; LID: Lidocaine hydrochloride; N.A.: Not available; NFZ: Nitrofurazone; PBS: Phosphate-buffered saline; PCL: Polycaprolactone; PDO: Polydioxanone; PeCL: Poly(ε-caprolactone); PEO: Polyethylene oxide; PHB: Polyhydroxybutyrate; PLA: Poly(l,d-lactic acid); PLLA: Poly(l-lactide); PVAc: Polyvinyl acetate; PVDF: Polyvinylidenefluoride; SA: Salicylic acid; SAG: Sodium alginate; SF: Silk fibroin; SS: Sericin; THY: Thymol; TRIS: Tris(hydroxymethyl)aminomethane; VE: Vitamin E derivative; ZN: Zein.

**Table 2 pharmaceutics-13-00183-t002:** Biological performance of the asymmetric membranes produced through electrospinning aimed to be used as wound dressings.

Composition	Antibacterial Activity	Antioxidant Activity	Cell Behavior	Wound Healing	Ref
PCL-PLA/GelMA-ChMA	N.A.	N.A.	Fibroblasts cellular viability of ≈101.1% and ≈106.7%, for the top and bottom layer, respectively, after 7 days; After seven days, a continuous layer of cells with typical fibroblastic morphology and lamellipodia connecting to surrounding mesh was observed	N.A.	[55]
CS-PCL/HA	The optical density of *E. coli* was ≈0.09 for asymmetric membrane and ≈0.37 for the control (PCL), after 24 h of incubation	N.A.	Vero cells viability of ≈147.52%, after 5 days; Vero cells adhered to the asymmetric membrane showed a good cell–cell interaction as well as an improved cell/fibrous scaffold integration, after 5 days	N.A.	[26]
CS/PDLLA	N.A.	N.A.	Efficient attachment of fibroblast cells to the membrane, after 5 days of incubation; Infiltration of fibroblast cells into CS/PDLLA membranes up to a depth of ≈32.5 μm	The histology analysis, after 7 days of the wound treatment with the asymmetric membrane, demonstrated that the epidermis and dermis layer were gradually restored with the successful regeneration of keratinocytes and fibroblasts, respectively	[64]
BGB/BGA	N.A.	N.A.	The proliferation of fibroblast cells on the top and bottom layer was ≈184.41% and ≈208.68%, respectively, after 7 days; The proliferation of keratinocytes on the top and bottom layer was ≈190.18% and ≈183.74%, respectively, after 7 days	The asymmetric membranes induced the reduction of the wound size in ≈83.1% after 14 days; The histology analysis of the wound covered with the asymmetric membrane showed the re-epithelialization and a structure resembling the normal skin, with skin-like organized collagen fibers	[61]
PCL-mupirocin_ CS-LID	The membrane showed excellent activity against *S. aureus* (35 mm of inhibition zone), *P. aeruginosa* (30 mm of inhibition zone), and *E. coli* (28 mm of inhibition zone)	N.A.	The relative cell number (OD) of the fibroblast cells was ≈0.61 after 7 days	N.A.	[68]
PeCL(nylon mesh pore size 40)/Gel-pio	The top layer exhibited lower bacterial adhesion in comparison to the control, against *S. aureus*, *E. coli*, and *P. aeruginosa*	N.A.	The viabilities of fibroblast cells and HUVECs were ≈179.38% and ≈353.85%, respectively, after 3 days; The percentages of fibroblast cells and HUVECs migration were 61.54% and 68.57%, respectively	Type 2 Diabetic Mice: The wounds treated with PCL40/Gel-pio were almost completely closed on day 10, whereas the other groups needed more than 14 days; The blood glucose concentrations of the PCL40/Gel-pio group were maintained at a low level and increased after day 7, while the others increased continuously over time; Completely regenerated epidermis and dense dermis, and continuous and uniform granulation tissue on day 14; The relative collagen content on day 14 was 59.52%; The asymmetric membranes group showed the highest density of newly formed blood vessels (≈30.32 mature vessels per field) after 7 days; The asymmetric membranes group showed the most potent effect on cell proliferation (higher Ki67 expression, ≈73.23 positive cells per field)	[54]
				Type 1 Diabetic Rat: Showed the fastest wound healing effect; The wounds treated with PCL40/Gel-pio were almost completely closed on day 14; Showed the higher density of collagen fibers (≈61.46%) after 7 days; Showed the highest density of newly formed blood vessels (≈32.27 vessels per field) after 5 days, and decreased to the day 14 (≈12.69 vessels per field); The asymmetric membranes group showed the most potent effect on cell proliferation (higher Ki67 expression, ≈56.55 positive cells per field), on day 14	
PLA-VE/PCL-VE	N.A.	N.A.	Fibroblast cells viability of ≈87.44%, after 10 days; The surface of the membrane was highly colonized by fibroblast cells, and the cells’ attachment inside the pores of the membranes was also observed, after 10 days	After 14 days, the chick chorioallantoic membrane assay revealed the complete coverage of the asymmetric membrane with the newly formed vessels (≈48.99 blood vessels)	[57]
PCL/PVAc-CRV (Sample I-PVAc in DMF/ETOH; sample II-PVAc in DMF)	Sample I inhibited the proliferation of *E. coli* (from 1.6 × 10^9^ to 1.2 × 10^7^ CFU/mL) and *S. aureus* (5.7 × 10^10^ to 2.3 × 10^7^ CFU/mL) Sample II inhibited the proliferation of *E. coli* (from 1.6 × 10^9^ to 1.4 × 10^6^) and *S. aureus* (5.7 × 10^10^ to 3.1 × 10^6^)	N.A.	The fibroblast cells viability was ≈108.24% and ≈145.29% for sample I and sample II, respectively, after 3 days After 3 days, fibroblast cells properly adhered and spread homogeneously on the membrane; No significant effects on cells migration in in vitro wound closure assays, after 3 days	N.A.	[66]
PCL-HA/CS-ZN-SA	The asymmetric membranes showed an inhibitory effect of ≈99% against *S. aureus* and presented an inhibitory halo of 9.84 ± 3.64 mm	N.A.	The fibroblast cells viability was ≈106.05%, after 7 days; After 7 days, the cells presented filopodia protrusions and were completely attached on both layers	N.A.	[28]
PCL/PEO-CS-AV	Low bacteria adhesion to the upper side of the top layer; The asymmetric membranes showed antibacterial activity of 99.99% and 99.97% against *S. aureus* and *E. coli*, respectively	N.A.	The fibroblast cells viability was ≈94.44%, after 7 days; After 3 days, the fibroblast cells attached and proliferated; The fibroblasts migrated to the interior of the asymmetric membrane (8–10 µm within the polymeric structure), after 3 days	N.A.	[67]
PCL-SF/SF-HA-THY	The PCL-SF layer avoided the bacterial infiltration of *S. aureus* and *P. aeruginosa*; The SF-HA-THY layer showed antibacterial activity of 87.42% (and an inhibition zone of ≈69.90%) and 58.43% (and an inhibition zone of ≈52.38%), against *S. aureus* and *P. aeruginosa,* respectively	Antioxidant activity of 9.22% and ≈45.64% for the PCL-SF and SF-HA-THY membranes, respectively, after 8 h of incubation	The fibroblast cells viability was ≈93.44% and ≈93.82% for the PCL-SF and SF-HA-THY membranes, respectively, after 7 days; Both membranes promoted the cell adhesion, but in the SF_HA_THY layer, the fibroblast cells appeared to present more filopodia protrusions, higher cell adhesion, and proliferation	N.A.	[20]
PeCL/PDO-TiO_2_ nanoparticles (concentration of 3% (PP3T5T) and 5% (PP5T5T))-TTC	PP3T5T presented an inhibition zone of 12.78 ± 2.5 and 16.28 ± 4.7 µm, against *S. aureus* and *E. coli*, respectively; PP5T5T presented an inhibition zone of 26.14 ± 6.7 and 36.94 ± 5.6 µm, against *S. aureus* and *E. coli*, respectively	N.A.	The fibroblast cells proliferation was ≈107.77% and ≈110.83% for PP3T5T and PP5T5T, respectively, after 6 days; The fibroblast cells penetrated up to a depth of 40 and 35 µm for PP3T5T and PP5T5T, respectively, after 4 days	N.A.	[69]
PLLA-SS (1:1, 2:1, 4:1)-NFZ (0.2%, 0.5%, 1.0%)/NFZ-PLLA	Inhibition zones of 20.41 ± 0.43 to 24.28 ± 0.10 mm against *E. coli* and 21.47 ± 0.19 to 27.04 ± 0.35 mm against *B. subtilis*	N.A.	The fibroblast cells viabilities were all above 95% for the PLLA-SS(2:1)–0.2% NFZ (with concentrations ranging from 10 to 2.5 mg × mL^−1^), after 3 days	The asymmetric membranes induced the reduction of the wound size in 97% after 12 days	[71]

AV: Aloe vera; BGA: β-glucan acetate; BGB: β-glucan butyrate; ChMA: Methacrylated chitosan; CRV: carvacrol; CS: Chitosan; Gel: Gelatin; GelMA: Methacrylated gelatin; HA: Hyaluronic acid; LID: Lidocaine hydrochloride; N.A.: Not available; NFZ: Nitrofurazone; PCL: Polycaprolactone; PDO: Polydioxanone; PeCL: Poly(ε-caprolactone); PEO: Polyethylene oxide; Pio: Pioglitazone; PLA: Poly(l,d-lactic acid); PLLA: Poly(l-lactide); PVAc: Polyvinyl acetate; SA: Salicylic acid; SF: Silk fibroin; SS: Sericin; THY: Thymol; VE: Vitamin E derivative; ZN: Zein.

**Table 3 pharmaceutics-13-00183-t003:** Asymmetric membranes produced through electrospinning aimed to be used as biomolecules delivery systems.

Aims	Composition	Biomolecule Incorporated	Layer of Incorporation	Encapsulation Efficiency and Loading Efficiency	Release Profile	Ref
Antibacterial activity	PCL/PVAc	CRV	Bottom layer	The CRV loaded in samples I and II was 3.0 ± 0.4 wt% and 2.3 ± 0.5 wt%, respectively	Sample I released about 45% of the total drug, while sample II released about 60% of the loaded CRV, at pH 8; after 7 days in basic pH the membranes were transferred to PBS pH 7.4 and after two weeks in this medium the release reached 60% and 85% of the loaded drug for samples I and II, respectively; after 14 days, the samples were put in an acidic medium where after one week the release reached 85% and 100% from the samples I and II, respectively	[66]
	(Sample I—PVAc in DMF/ETOH; sample II—PVAc in DMF)			Encapsulation efficiencies were 55 ± 5% and 43 ± 9% for samples I and II, respectively		
	PCL-HA/CS-ZN	SA	Bottom layer	N.A.	The release profile, in PBS (pH 5.5), consisted of a burst release in the first hour followed by a sustained release for 5 days (reaching approximately 16%)	[28]
	PCL/PEO-CS	AV	Bottom layer	N.A.	N.A.	[67]
	PCL-SF/SF-HA	THY	Bottom layer	Encapsulation efficiency of 79.7 ± 7.19%	THY release from the nanofibers, at both pH levels, comprises a burst release in the first 8 h after immersion in PBS, followed by a gradual release up to 24 h	[20]
				Loading efficiency of 64.8 ± 5.42%	At pH 8, the release of THY reached a maximum of 91.87 ± 0.99%	
					At pH 5, the release of THY reached a maximum of 71.75 ± 2.06%	
	PeCL/PDO	TiO_2_ nanoparticles (concentration of 3% (PP3T5T) and 5% (PP5T5T)) and TTC	Bottom layer	N.A.	The release profile of TTC, in PBS (pH 7.4), from PP3T5T showed an initial burst release of 47.2% within the first 6 h, followed by a slow release that reached 61.9% until day 4	[69]
					The burst release of TTC, in PBS (pH 7.4), from PPT5T5 was 50.8% within the first 6 h and reached 77% over 4 days	
	PLLA-SS/PLLA	NFZ	Both layers	N.A.	The top PLLA-SS nanofibrous mats with 0.2% of NFZ, in PBS (pH 7.4) presented a fast release profile with more than 98% of NFZ detected in 10 min of incubation for every ratio	[71]
					The PLLA bottom layer in PBS (pH 7.4) presented a more controlled and sustained release, reaching 17.6% after 48 h	
					PLLA-SS(2:1)-0.2NFZ/PLLA-2NFZ, PLLA-SS(2:1)-0.5NFZ/PLLA-2NFZ, and PLLA-SS(2:1)-1.0NFZ/PLLA-2NFZ in PBS (pH 7.4) presented a burst release of 11.2%, 14.3%, and 28.4%, respectively, and the release amounts reached 29.4%, 43.0%, and 53.9%, respectively, after 48 h	
Wound healing improvement	PCL/CS	Mupirocin	Top layer	N.A.	The initial burst release of LID reached 66% in the first hours and increased gradually to 85% in the following 6 h, in PBS	[68]
		LID	Bottom layer		The release of mupirocin consisted in the release of 57% of mupirocin in the first 6 h, followed by a sustained release (30% was released in the following 114 h), in PBS	
	PeCL/Gel	Pio	Bottom layer	Loading efficiency of 56.16 ± 7.45%	The Pio release rapidly reached 40% in day 1 and a long-term release reached 75% in day 14, in PBS (pH 7.4)	[54]
	PLA/PCL	VE	Both layers	N.A.	The asymmetric membrane showed a sustained release of VE over 21 days reaching a maximum of 78%, in PBS	[57]

AV: Aloe vera; CRV: Carvacrol; CS: Chitosan; Gel: Gelatin; HA: Hyaluronic acid; LID: Lidocaine hydrochloride; N.A.: Not available; NFZ: Nitrofurazone; PBS: Phosphate-buffered saline; PCL: Polycaprolactone; PDO: Polydioxanone; PeCL: Poly(ε-caprolactone); PEO: Polyethylene oxide; Pio: Pioglitazone; PLA: Poly(l,d-lactic acid); PLLA: Poly(l-lactide); PVAc: Polyvinyl acetate; SA: Salicylic acid; SF: Silk fibroin; SS: Sericin; THY: Thymol; VE: Vitamin E derivative; ZN: Zein.

## Data Availability

No new data were created or analyzed in this study. Data sharing is not applicable to this article.

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
