# Peer review of "Electrospun Asymmetric Membranes as Promising Wound Dressings: A Review"

_pharmaceutics, 2021, doi:10.3390/pharmaceutics13020183_

Round 1

Reviewer 1 Report

This review summarizes the latest research progress of electrospun asymmetric membranes as promising wound dressings. The authors summarized the research progress of the asymmetric membranes from various aspects such as asymmetric membranes preparation method, asymmetric membranes type, and asymmetric membranes as drug delivery systems. Although the entry point of this article is very narrow, but the author's summary is very comprehensive and systematic, so I think this article can be published in pharmaceutics after the following revisions:

  1. The introduction part of this article should be reconstructed. The author should focus on the advantages of the asymmetric membranes compared with other wound dressings, so that readers can understand the significance of this review.
  2. The pictures selected in review should be very representative, so I think the author should re-select the pictures in the article. For example, I think Figure 3 is not representative.
  3. Examples of reviews should not be simple lists, so the author should add more discussion and summary after the examples in this review.
  4. Conclusions and Future Perspectives are the key part of a review. The author should add more detailed future perspectives in this article, and the future perspectives should start from existing research.

Author Response

The authors would like to acknowledge the reviewers for their comments, which were fundamental to enhance the impact and quality of this manuscript. Further, the manuscript was carefully revised and all the recommended modifications were performed. Therefore, the authors trust that the revised version of the manuscript is now suitable for publication in the Pharmaceutics Special Issue on “Wound Care: From Traditional Cotton Wound Dressings to Innovative Wound Dressings”.

Reviewer 1:

This review summarizes the latest research progress of electrospun asymmetric membranes as promising wound dressings. The authors summarized the research progress of the asymmetric membranes from various aspects such as asymmetric membranes preparation method, asymmetric membranes type, and asymmetric membranes as drug delivery systems. Although the entry point of this article is very narrow, but the author's summary is very comprehensive and systematic, so I think this article can be published in pharmaceutics after the following revisions:

  • The introduction part of this article should be reconstructed. The author should focus on the advantages of the asymmetric membranes compared with other wound dressings, so that readers can understand the significance of this review.

The authors acknowledge the reviewer comment. The introduction section was modified to clarify the need to develop novel and more effective wound dressings. The advantages of asymmetric membranes were further described in topic 2. Asymmetric Membranes.

  • The pictures selected in review should be very representative, so I think the author should re-select the pictures in the article. For example, I think Figure 3 is not representative.

According to the reviewer’s suggestion, new figures were selected and added to the manuscript to provide a better representation of the asymmetric membranes’ application in the wound healing.

  • Examples of reviews should not be simple lists, so the author should add more discussion and summary after the examples in this review.

The authors acknowledge the reviewer’s suggestion, the manuscript was modified to provide a more insightful discussion about the different works cited in the text. Nevertheless, in the time provided to perform the revisions, it was not possible to perform extensive text and structural modifications.

  • Conclusions and Future Perspectives are the key part of a review. The author should add more detailed future perspectives in this article, and the future perspectives should start from existing research.

Considering the different comments regarding the Conclusions and future perspectives, this section was revised to provide additional information about the asymmetric membranes current stage of clinical use and future development to achieve complete skin regeneration.

Reviewer 2 Report

  • The authors begins the abstract by describing limitations in skin substitutes then describe wound dressings. Skin substitutes and wound dressings are not the same. In most contexts, skin substitutes integrate into the host wound/skin and do not requires changes or removal after healing, while wound dressings require changes and removal after wound closure. The authors need to distinguish this or explain more clearly in the abstract what the review is about.
  • Similarly, the introduction section lacks a review and limitations of wound dressings. The authors again focus on skin substitutes. The authors need to mention current wound dressings used in the clinic that can be classified in the same category as asymmetric membranes as opposed to bioengineered skin substitutes. The authors use the terms 'skin substitute' and 'wound dressings' synonymously which is not true. If they are reviewing asymmetric membranes as skin substitutes then it should be clearly defined.
  • The authors begin describing the methods for producing asymmetric membranes without defining what are asymmetric membranes. A definition or clear description should be included.
  • Please describe if the membranes produced using the methods in section 2 can be used to incorporate cells in them. If not, describe it as a limitation as such. The authors only describe that the membranes are conducive to cell infiltration and proliferation but do not describe if cells can be incorporated in the membranes using the production methods for clinical use. Again, the reader gets the impression that asymmetric membranes are wound dressings and not skin substitutes.
  • In section 3, please describe parameters either of polymer composition or electrospinning set up that influence drug loading and release profiles.
  • Figure 4 should be edited to clearly show the utility of drug/therapeutic agents loaded asymmetric membranes in wound healing.
  • Within the sections the authors should desciibe ideal or most biocompatible materials/polymers for asymmetric membranes for wound healing.
  • In the conclusion/furture perspective section, the authors should describe the current stage of clinical use (clinical trials or FDA approvals, etc) for asymmetric membranes for wound healing applications.

Author Response

The authors would like to acknowledge the reviewers for their comments, which were fundamental to enhance the impact and quality of this manuscript. Further, the manuscript was carefully revised and all the recommended modifications were performed. Therefore, the authors trust that the revised version of the manuscript is now suitable for publication in the Pharmaceutics Special Issue on “Wound Care: From Traditional Cotton Wound Dressings to Innovative Wound Dressings”.

Reviewer 2:

  • The authors begins the abstract by describing limitations in skin substitutes then describe wound dressings. Skin substitutes and wound dressings are not the same. In most contexts, skin substitutes integrate into the host wound/skin and do not requires changes or removal after healing, while wound dressings require changes and removal after wound closure. The authors need to distinguish this or explain more clearly in the abstract what the review is about.

The authors acknowledge the reviewer for raising this important concern. This review focus in the application of asymmetric electrospun membranes as wound dressings. Therefore, the manuscript was modified to increase clarity and avoid any possible confusions between wound dressing and skin substitutes.

  • Similarly, the introduction section lacks a review and limitations of wound dressings. The authors again focus on skin substitutes. The authors need to mention current wound dressings used in the clinic that can be classified in the same category as asymmetric membranes as opposed to bioengineered skin substitutes. The authors use the terms 'skin substitute' and 'wound dressings' synonymously which is not true. If they are reviewing asymmetric membranes as skin substitutes then it should be clearly defined.

The manuscript was revised accordingly the reviewer’s comment to clarify that this review focus on the application of asymmetric electrospun membranes as wound dressings.

  • The authors begin describing the methods for producing asymmetric membranes without defining what are asymmetric membranes. A definition or clear description should be included.

The authors acknowledge the reviewer's comment. A definition of asymmetric membranes was added to the topic 2 “Asymmetric membranes are 3D matrices composed of two layers, which enable a high similarity with the native skin, while protecting the wound against bacterial penetration, dehydration, and exudate accumulation.”.

  • Please describe if the membranes produced using the methods in section 2 can be used to incorporate cells in them. If not, describe it as a limitation as such. The authors only describe that the membranes are conducive to cell infiltration and proliferation but do not describe if cells can be incorporated in the membranes using the production methods for clinical use. Again, the reader gets the impression that asymmetric membranes are wound dressings and not skin substitutes.

Authors acknowledge the reviewer for this comment. The capacity to incorporate cell during the production of asymmetric membranes can be highly important to enhance the wound healing, namely, to achieve the complete regeneration of the skin and its appendages. In our research, considering the different methods described in section 2, we only found works reporting the cell incorporation during the production process for the electrospinning technique. Therefore, this capacity was highlighted in section 2.1. “Moreover, the electrospinning technique can also be optimized to be compatible with the cell encapsulation enhancing the wound healing capacity, contrasting with the other techniques explored for producing asymmetric polymeric membranes.”.

  • In section 3, please describe parameters either of polymer composition or electrospinning set up that influence drug loading and release profiles.

According to the reviewer’ suggestion, the influence of the polymer composition and electrospinning set up on the drug loading and release profiles was added to the revised version of the manuscript.

  • Figure 4 should be edited to clearly show the utility of drug/therapeutic agents loaded asymmetric membranes in wound healing.

The authors acknowledge the reviewer's comment. Figure 4 was carefully revised to show the main mechanisms behind the biomolecules’ antibacterial activity and wound healing enhancer capacity.

  • Within the sections the authors should describe ideal or most biocompatible materials/polymers for asymmetric membranes for wound healing.

According to the reviewer’ suggestion, a new section (2.1.1. Polymers) was added to the manuscript, describing the polymers usually selected for producing fibrous mats for wound healing application.

  • In the conclusion/future perspective section, the authors should describe the current stage of clinical use (clinical trials or FDA approvals, etc) for asymmetric membranes for wound healing applications.

The authors acknowledge the reviewer for his comments. However, the authors did not find any clinical trial or FDA approval for the asymmetric membranes’ application as wound dressings. In this way, the potentiality of the asymmetric structure to enhance skin regeneration is demonstrated with the reference of the dermal/epidermal skin substitutes used already in the clinic.

Reviewer 3 Report

Dear Editor of Pharmaceutics Journal, 

I finished the peer review of the manuscript pharmaceutics-1058515 entitled “Electrospun asymmetric membranes as promising wound dressings: a review.” 

The manuscript is well written and represents well-summarized updated information regarding the advantages of asymmetric membranes synthesized by electrospun methods for tissue engineering and wound dressings. I found the review manuscript helpful. 

Based on this, I consider this review provides well-organized updated information regarding the importance of electrospun asymmetric membranes and their usage in wound dressings. Thus, I recommend to the Editor to publish it after considered the fulfill of the following comments to improve the quality of the manuscript:

  1. Arrange the data from table 1 logically. For example, use the composition of the top layer to present the data. Therefore the readers could make some assumptions regarding the properties of the membranes and their components. Try to find and present a logical pattern to organize the summarized data. 
  2. Dense top layers are duplicated in figure 2; please correct and write the proper names.
  3. Add another table to note the biological activity of the asymmetric membranes produced through electrospinning used as wound dressing. (e.g., bacterial adhesion, in vitro or in vivo cell proliferation and differentiation, wound healing closure, among others). Include the data reported on page 6, lines 215 to 243. This table should reflect the content described in the title of the manuscript.
  4. In figure 3, add a picture of the PCL membrane.
  5. To allow the readers to compare the benefits of using electrospun asymmetric membranes as drug delivery systems. Use a table to summarize the presented information. Order it logically to allow reasonable comparisons between each report, especially those related to antibacterial properties and wound-healing process.
  6. The topic written under the title of section 3 is broad. Therefore, using the terminology “drug” to describe it would not reflect the vast content. The authors provide examples of asymmetric membranes used for drug delivery and natural products such as essential oils. Please consider renaming this section; biomolecules or bioactive molecules could be an assertive word to be used herein.
  7. Improve the quality of plots in figures 3, and 6b

Author Response

The authors would like to acknowledge the reviewers for their comments, which were fundamental to enhance the impact and quality of this manuscript. Further, the manuscript was carefully revised and all the recommended modifications were performed. Therefore, the authors trust that the revised version of the manuscript is now suitable for publication in the Pharmaceutics Special Issue on “Wound Care: From Traditional Cotton Wound Dressings to Innovative Wound Dressings”.

Reviewer 3: The manuscript is well written and represents well-summarized updated information regarding the advantages of asymmetric membranes synthesized by electrospun methods for tissue engineering and wound dressings. I found the review manuscript helpful. Based on this, I consider this review provides well-organized updated information regarding the importance of electrospun asymmetric membranes and their usage in wound dressings. Thus, I recommend to the Editor to publish it after considered the fulfill of the following comments to improve the quality of the manuscript:

  • Arrange the data from table 1 logically. For example, use the composition of the top layer to present the data. Therefore the readers could make some assumptions regarding the properties of the membranes and their components. Try to find and present a logical pattern to organize the summarized data.

According to the reviewer suggestion, Table 1 was organized according to the main material used to produce the top layer (please see Table 1, page XX, lines XX).

  • Dense top layers are duplicated in figure 2; please correct and write the proper names.

Figure 2 was corrected according to the reviewer’s comment.

  • Add another table to note the biological activity of the asymmetric membranes produced through electrospinning used as wound dressing. (e.g., bacterial adhesion, in vitro or in vivo cell proliferation and differentiation, wound healing closure, among others). Include the data reported on page 6, lines 215 to 243. This table should reflect the content described in the title of the manuscript.

According to the reviewer suggestion, a table describing the biological activity of the asymmetric electrospun membranes used as wound dressings was added to the revised version of the manuscript (please see Table 2, page XX, lines XX).

  • In figure 3, add a picture of the PCL membrane.

The authors acknowledge the reviewer for this suggestion. SEM images of the PCL membrane were added to Figure 3 (please see page XX, lines XX).

  • To allow the readers to compare the benefits of using electrospun asymmetric membranes as drug delivery systems. Use a table to summarize the presented information. Order it logically to allow reasonable comparisons between each report, especially those related to antibacterial properties and wound-healing process.

The authors acknowledge the reviewer for raising this important concern. According to the reviewer suggestion, a table summarizing the application of electrospun asymmetric membranes as drug delivery systems was added to the revised version of the manuscript (please see Table 3, page XX, lines XX). Moreover, since the biological performance of the asymmetric electrospun membranes is already described in Table 2, this information was not included in Table 3.

  • The topic written under the title of section 3 is broad. Therefore, using the terminology “drug” to describe it would not reflect the vast content. The authors provide examples of asymmetric membranes used for drug delivery and natural products such as essential oils. Please consider renaming this section; biomolecules or bioactive molecules could be an assertive word to be used herein.

According to the reviewer suggestion, the title of section 3 was revised for “Electrospun asymmetric membranes as delivery systems of biomolecules” (please see page XX, line XX).

  • Improve the quality of plots in figures 3, and 6b

The authors acknowledge the reviewer comment, figure 3 and 6 were modified to be more representative of the asymmetric membranes. Nevertheless, the plots presented in figures 3 and 6b are reproduced from published articles and we do not have permissions for making changes on the figures.

Round 2

Reviewer 2 Report

The authors have adequately addressed all comments and have included several details and sections to further improve the impact of this review.

The authors are requested to prove read the manuscript once to ensure no sentences are repeated as it appears not with track changes on.